# CAuSE: Post-hoc Natural Language Explanation of multimodal classifiers through causal abstraction

## Abstract

The increasing integration of AI models in critical areas, such as healthcare, finance, and security has raised concerns about their "black-box" nature, limiting trust and accountability. To ensure robust and trustworthy AI, interpretability is essential. In this paper, we propose **CAuSE (Causal Abstraction under Simulated Explanation)**, a novel framework for post-hoc explanation of multimodal classifiers. Unlike existing interpretability methods, such as Amnesic Probing and Integrated Gradients, CAuSE generates causally faithful natural language explanations of fine-tuned multimodal classifiers' decisions. CAuSE integrates Interchange Intervention Training (IIT) within a Language Model (LM) based module to simulate the causal reasoning behind a classifier's outputs. We introduce a novel metric *Counterfactual F1 score* to measure causal faithfulness and demonstrate that CAuSE achieves state-of-the-art performance on this metric. We also provide a rigorous theoretical underpinning for causal abstraction between two neural networks and implement this within our CAuSE framework. This ensures that CAuSE's natural language explanations are not only simulations of the classifier's behavior but also reflect its underlying causal processes. Our method is task-agnostic and achieves state-of-the-art results on benchmark multimodal classification datasets, such as e-SNLI-VE and Facebook Hateful Memes, offering a scalable, faithful solution for interpretability in multimodal classifiers.

## 1 Introduction

With the rise of Visual Language Models (VLMs), AI systems have evolved to handle multiple data types like images, text, and audio. Multimodal classifiers, central to this advancement, are crucial in applications such as healthcare, where they combine medical images and patient data to improve diagnostic accuracy for diseases like COVID-19 and Alzheimer's (Baltrušaitis et al., 2017). Similarly, in autonomous driving, they enhance decision-making by integrating visual, LiDAR, and radar inputs (Xiao et al., 2022). These classifiers boost performance by leveraging diverse modalities, making them vital in real-world scenarios.

However, as multimodal classifiers grow in complexity, the need for interpretability becomes paramount. Current interpretability methods, such as Integrated Gradients(Sundararajan et al., 2017a), are designed to highlight explicit input features but fall short of capturing the implicit causal relationships that often drive the decisions of these models. While some techniques, like CausaLM(Feder et al., 2022) and Amnesic Probing(Elazar et al., 2021), aim to incorporate causal mechanisms for interpretability, they struggle with scalability. Other methods, such as Semantify(Bandyopadhyay et al., 2024), manage implicit concepts efficiently but are restricted to specific use cases and fail to generate comprehensive natural language explanations.

To address these limitations, large Visual Language Models (VLMs) have been utilized to generate natural language explanations for decisions made by visual-text multimodal classifiers. However, these models often inject their own biases and opinions, leading to explanations that are inconsistent or detached from the actual workings of the classifier(Agarwal et al., 2024). Recent studies(Madsen et al., 2024) have highlighted these faithfulness issues, revealing inconsistencies when models are further probed.

In this paper, we introduce **CAuSE (Causal Abstraction under Simulated Explanation)**, a novel framework designed to generate faithful natural language explanations for the decisions of a pre-trained classifier, offering post-hoc interpretability. CAuSE combines Interchange Intervention Training(Geiger et al., 2021a) with Language Model (LM)-based modules, ensuring that the generated explanations are both causally accurate and reflective of the classifier's internal decision-making process. We introduce a new metric, the *Counterfactual F1 score*, to assess the causal faithfulness of explanations. CAuSE sets a new benchmark on this metric, achieving state-of-the-art performance. Through case studies, we showcase successful generations from our framework and conduct error analysis to identify common mistakes and their underlying causes.

Our framework is task-agnostic and demonstrates state-of-the-art performance on benchmark datasets, such as e-SNLI-VE(Do et al., 2021) and Facebook Hateful Memes(Kiela et al., 2021), providing robust, faithful explanations across diverse multimodal tasks. The codes are available at `https://anonymous.4open.science/r/CAuSE-5BD0`.

## 2 ARCHITECTURE

Our framework, CAuSE, generates faithful natural language explanations for decisions made by a pre-trained multimodal classifier (called the **post-hoc classifier**). As detailed in Section 3.2, CAuSE acts as a causal abstraction of the post-hoc classifier, ensuring its explanations are rooted in the actual decision-making process. This is supported by the high Counterfactual F1 scores CAuSE achieves compared to the other ablated components, as shown in Table 2. This section introduces the post-hoc classifier and provides a detailed description of the CAuSE framework, with a working diagram of both presented in Figure 1.

### 2.1 POST-HOC CLASSIFIER

The post-hoc classifier is assumed to be composed of a multimodal encoder $E$ and a feed-forward neural network (FFN) $C_1$.

**Multimodal Encoder.** The multimodal encoder $E$ accepts as inputs the text ($t \in \mathbb{R}^{m \times 1}$) and image representation ($v \in \mathbb{R}^{m \times 1}$). The image and text representation are fused via either i) early-fusion or ii) late-fusion modules. The final multimodal representation is denoted as $c \in \mathbb{R}^{m \times 1}$, where $c = E(t, v)$.

This module serves as a plug-and-play replacement for any multimodal encoder, whether based on early-fusion or late-fusion. In our implementation for this paper, we use a late-fusion-based module, which consists of CLIP(Radford et al., 2021) and MFB(Yu et al., 2017), as commonly adopted in the literature(Bandyopadhyay et al., 2024).

**Classifier $C_1$.** The classifier gets the multimodal representation $c$ and via a chain of feed-forward neural nets, it gets transformed into a vector $z \in \mathbb{R}^{L \times 1}$, where $L$ is the number of classes in the output label. A softmax function is used which converts logit $z$ into a probability distribution $y_1 = softmax(z)$. Supposing the one-hot ground truth probability distribution is $\hat{y}_1$, the cross-entropy loss which is used to optimize the post-hoc classifier is

$$L_{PH} = -[\hat{y}_1 log(y_1)] \tag{1}$$

### 2.2 CAuSE

The CAuSE is composed of i) A language model (LM) called $\phi_1$ which reconstructs the input text. ii) Another LM $\phi_2$ which generates the explanation. $\phi_2$ is coupled with another classifier ($C_2$) which is trained to predict the outputs of the original classifier $C_1$. *It is important to note that $\phi_1$ and $\phi_2$ share the same weights and are both implemented using a single GPT-2 small model with 350 million parameters, reducing memory consumption.*

**Training the LMs.** The LMs are trained using vanilla causal language modelling (CLM) loss. Specifically, the multimodal representation $c$ is broken into two components $c_0$ and $c_1$ by passing them through two separate FFNs ($F_0$ and $F_1$) which bring their dimension to match with LM embedding dimension $\mathbb{R}^{768 \times 1}$, such that $c_0 = F_0(c)$, and $c_1 = F_1(c)$.

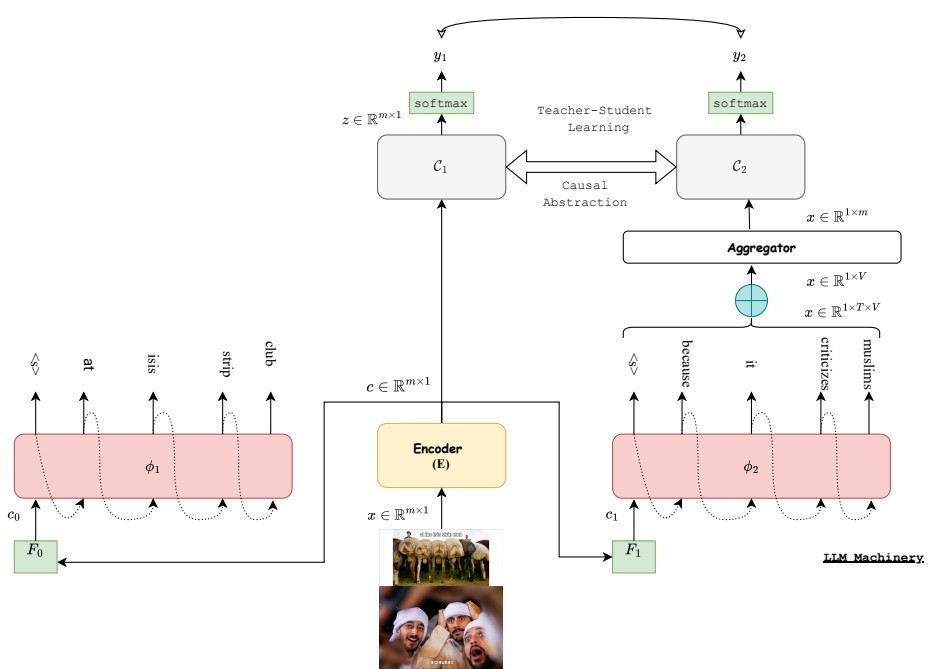

Figure 1: Diagram of our proposed framework CAuSE and the post-hoc classifier.

Given $c_0$, $\phi_1$ reconstructs next word $(x_i)$ for the $i$-th step via the following loss over a total of $T'$ time-steps:

$$\mathcal{L}_{\phi_1} = -\sum_{i=1}^{T'} log P_{\phi_1}(x_i|x_{i-1}) \quad \text{where} \quad x_0 = c_0 \tag{2}$$

Similar equation is used to train $\phi_2$

$$\mathcal{L}_{\phi_2} = -\sum_{i=1}^{T} log P_{\phi_2}(x_i|x_{i-1}) \quad \text{where} \quad x_0 = c_1 \tag{3}$$

**Aggregator** $A$. The logits $x_i$ retrieved from $\phi_2$ has the dimension $\mathbb{R}^{1 \times T \times V}$, where $V$ is the vocabulary size. These logits are first summed up along the time axis, which yields an intermediate vector $x$ having dimension of $\mathbb{R}^{1 \times V}$. This is then passed through another FFN which converts into a dimension same as $c$, which is $\mathbb{R}^{m \times 1}$.

**Classifier** $\mathcal{C}_2$. The aggregated output having the same dimension as $c$ is passed through a classifier $\mathcal{C}_2$ architecturally identical to $\mathcal{C}_1$. $\mathcal{C}_2$ is then trained to predict labels from $\mathcal{C}_1$[1]. $y_1$ is the output distribution from $\mathcal{C}_1$. Similarly, the probability distribution of $\mathcal{C}_2$ is $y_2 = softmax(\mathcal{C}_2(x))$, where $x = (A \circ \phi_2 \circ F_1)(c)$. We minimize the Cross-Entropy loss between outputs of $\mathcal{C}_2$ and $\mathcal{C}_1$ as:

$$\mathcal{L}_C = -[y_1 log(y_2)] \tag{4}$$

## 3 TRAINING METHODOLOGY

Training CAuSE involves two steps other than using $\mathcal{L}_C$ to align $\mathcal{C}_2$ to $\mathcal{C}_1$. They are i) Linguistic Infusion, ii) Causal Intervention.

---

[1]because we want to **mimic** $\mathcal{C}_2$ using $\mathcal{C}_1$ output.

## 3.1 LINGUISTIC INFUSION (LI)

We denote the input to the classifier $C_1$ as $c$, which is a multimodal encoding from the encoder. This captures the overall encoded representation of the multimodal source input. Through LI, we want to enrich $c$ with input source $(t, v)$ such that the latter could possess enough source information. LI is performed because: We only use a projected version of $c$ as the input token representation $c_2$ to $\phi_2$. This essentially serves as a bottleneck and most of the source information is lost when input is given to the LLM.

Assuming $M = (t, v)$, in LI, the enrichment of $c$ through source can be defined as the following constrained maximization problem following Plug and Play Language Model (PPLM)(Dathathri et al., 2020).

$$\hat{c} = \arg\max_c P(c|M) \quad \text{such that} \quad C_1(\hat{c}) = C_1(c) \tag{5}$$

Applying Bayes' theorem, $P(c|M) \propto P(c)P(M|c)$. Subsequently, the optimization Equation 5 can be written as: $\hat{c} = \arg\max_c P(M|c)$.

To estimate $P(M|c)$, we use an autoencoder which tries to predict $M$ from $c$. Formally, we try to estimate $P(d|c)$ by training an autoencoder which is trained to minimize a loss denoted by $L_{AE} = |d - M|$. This ensures $d$ becomes as close to $M$ as possible. Specifically, to find $\hat{c}$, we train the autoencoder first and then perform gradient descent of $c$ along the loss. We use $\hat{c} \leftarrow c - \gamma \nabla_c L_{AE}$ as the iterative update formula to get $\hat{c}$ from $c$.

## 3.2 CAUSAL INTERVENTION

**Causal Abstraction.** In Geiger et al. (2021c), the authors introduced the concept of causal abstraction for neural models. They define a neural network, $N_2$, as a causal abstraction of a higher-level causal model, $N_1$, if the neural representations of $N_2$ exhibit the same causal properties as the corresponding high-level variables in $N_1$. This alignment is achieved through the Interchange Intervention Training (IIT) objective.

A natural extension of this idea is to consider $N_1$ as a structurally identical neural network to $N_2$ and apply IIT between them, keeping $N_1$ frozen. This process ensures that $N_2$ becomes a causal abstraction of $N_1$. In our framework, we replace $N_1$ with $C_1$ and $N_2$ with $C_2$. Through IIT, we aim to ensure that the structurally identical classifier $C_2$ becomes a causal abstraction of $C_1$.

**Benefits of Causal Abstraction.** The type of causal abstraction learned through IIT is referred to as *constructive abstraction* in the causality literature. This concept ensures a systematic correspondence between interventions on the neurons in $N_1$ and those in $N_2$. Unlike a traditional teacher-student loss, which merely teaches the student to mimic the teacher's output, causal abstraction ensures that the student model internally mirrors the teacher's decision-making process. Through IIT, we guarantee that interventions on $N_1$ have corresponding effects on $N_2$, meaning that $N_2$ operates in the same causal manner as $N_1$.

We theoretically demonstrate that applying IIT can have significant implications if specific conditions are met. *Notably, when the weights of $C_1$ and $C_2$ remain the same throughout the IIT process*:

- The *LLM machinery* (i.e., $A$, $\phi_2$ along with $F_1$, combined as $F(z) = (A \circ \phi_2 \circ F_1)(z)$) perfectly simulates the encoder, such that for any input $x$, $F(E(x)) = E(x)$. Hence, the output from the LLM machinery matches that of the encoder [proven in **Theorem 1**].
- Building on this result, under a specific set of assumptions, we further show that the LLM machinery, together with $C_2$ (referred to as the *"explanator"*), forms a causal abstraction of the encoder and $C_1$ (the *"post-hoc classifier"*) [proven in **Theorem 2**].

**Teacher-student objective.** Figure 2 illustrates the training process for $C_2$. A sample input, consisting of both an image and a text from the dataset, is passed through the encoder. The encoder produces an output $c$, represented as a 3-dimensional vector, which is then fed into $C_1$. Assuming the weights in the first layer are all set to one, the activation of the $i_1$-th neuron (as shown in the diagram) would be calculated as $1 \times 0.1 + 1 \times 0.2 + 1 \times 0.3 = 0.6$. The final activation is then computed as $y_1 = 3 \times 0.6 + 2 \times 0.6 = 3$.

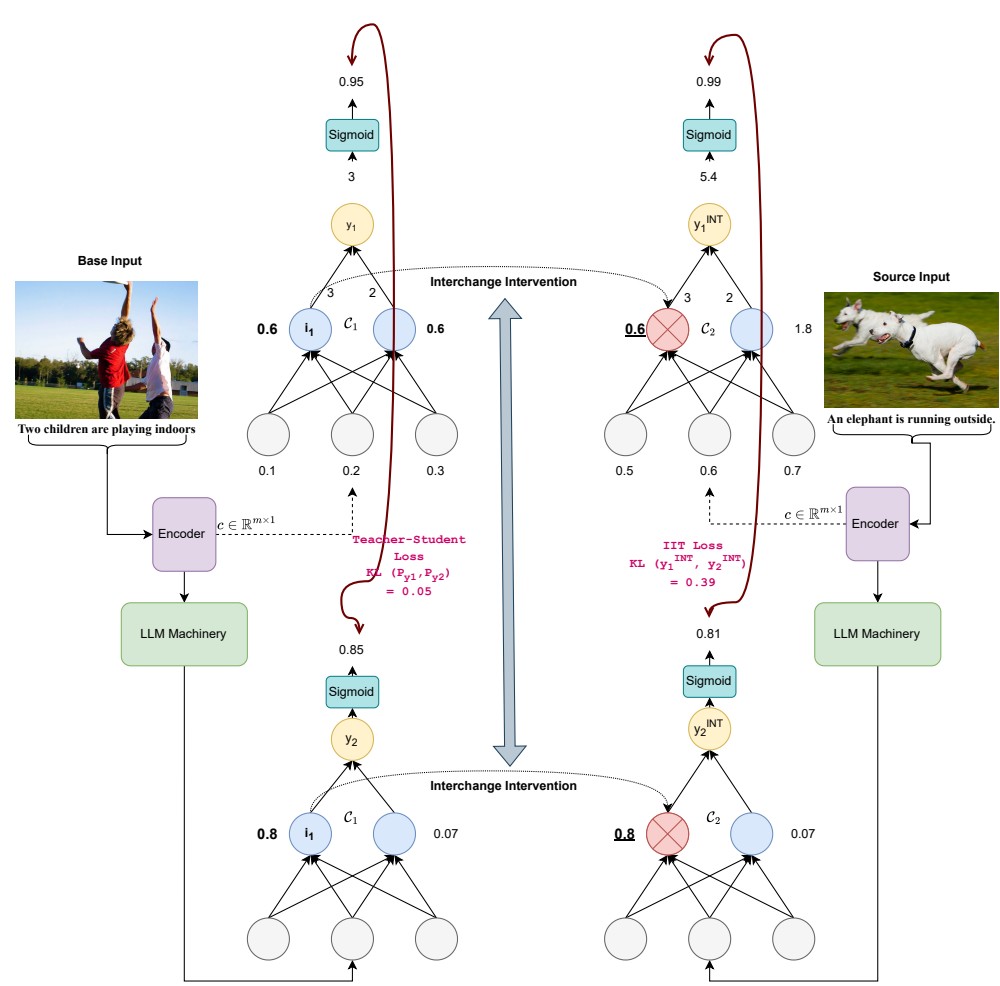

Figure 2: Causal Abstraction is enabled by IIT objective. Along with the teacher-student training objective, IIT poses as indispensable for $\mathcal{C}_2$ to be a causal abstraction of $\mathcal{C}_1$.

Simultaneously, the output $c$ is passed through the LLM machinery, which generates an activation that is forwarded to $\mathcal{C}_2$, producing an activation denoted as $y_2$. To ensure $\mathcal{C}_2$ mirrors the behavior of $\mathcal{C}_1$, we calculate the final loss using the KL divergence between their outputs:

$$\mathcal{L}_{TS} = KL(P_{y_1}|P_{y_2}) \tag{6}$$

where $P_{y_1} = [\sigma(y_1), 1 - \sigma(y_1)]$ and $P_{y_2} = [\sigma(y_2), 1 - \sigma(y_2)]$. This approach can be generalized to handle multiple outputs by applying the softmax function.

**IIT objective.** The Interchange Intervention (II) process is depicted in Figure 2. A neuron is randomly selected from $\mathcal{C}_1$ (denoted as $i_1$), and the II is applied. For a given source input, let $c = [0.5, 0.6, 0.7]$ (shown on the right-hand side). The II process ensures that the value of neuron $i_1$ is replaced with its original value, 0.6, which was obtained when the base input was processed. The final value after this intervention, referred to as the "intervened output," is represented as $y_1^{INT}$ for $\mathcal{C}_1$.

The same operation is carried out for $\mathcal{C}_2$, and the resulting "intervened output" is denoted as $y_2^{INT}$. Following the methodology of Geiger et al. (2021c), to ensure that $\mathcal{C}_2$ becomes a causal abstraction of $\mathcal{C}_1$, we minimize the IIT loss between the two outputs:

$$\mathcal{L}_{IIT} = KL(P_{y_1^{INT}}|P_{y_2^{INT}}) \tag{7}$$

**CAuSE Loss Function.** The final loss used to train CAuSE (i.e. $\mathcal{L}_{CAuSE}$) is defined as a sum of all individual loss terms.

$$\mathcal{L}_{CAuSE} = \mathcal{L}_{\phi_1} + \mathcal{L}_{\phi_2} + \mathcal{L}_{IIT} + \mathcal{L}_{TS} + \mathcal{L}_C + \|W_{\mathcal{C}_1} - W_{\mathcal{C}_2}\|_F \tag{8}$$

where $\|W_{\mathcal{C}_1} - W_{\mathcal{C}_2}\|_F$ denotes the frobenius norm between the weights of $\mathcal{C}_1$ and $\mathcal{C}_2$ respectively. *This term ensures that weights of $\mathcal{C}_1$ and $\mathcal{C}_2$ remain the same during training.*

**Counterfactual F1 score.** We hypothesize that if the explanator becomes a causal abstraction of the post-hoc classifier, it should still mimic the classifier under counterfactual input. To evaluate this, we introduce the counterfactual F1 (c-F1) score. Our empirical analysis shows that using only teacher-student training results in poor performance on counterfactual input, as reflected by a low c-F1 score. However, when combined with IIT, the explanator achieves a robust c-F1 score. Algorithm 1 details c-F1 calculation, and Table 2 compares methods based on their c-F1 scores.

### 3.2.1 Calculating Counterfactual F1 score

Suppose $x \in \mathcal{T}$ is a data-point from test set. As posed in Feder et al. (2022), the corresponding counterfactual input $x'$ for the post-hoc classifier would satisfy the following:

$$x' = \arg\min_{x' \in \mathcal{T}} d(x, x') \quad \text{such that} \quad \mathcal{C}_1(x) \neq \mathcal{C}_1(x') \tag{9}$$

$d$ is any kind of distance metric (e.g. manhattan, euclidean etc) between these data points. $\mathcal{C}_1(z)$ denotes the output class from $\mathcal{C}_1$ for any input $z$.

Subsequently, any counterfactual for $x$ can be expressed as: $x' = x + \mu$, where $\mu = x' - x$ is the perturbation between normal and counterfactual input. Note that $E(x')$ could not be a good counterfactual input for the LLM machinery, as $x' \in \mathcal{T}$ and high simulation performance between $\mathcal{C}_2$ and $\mathcal{C}_1$ means $\mathcal{C}_2$ could easily find label of $x'$. Therefore, we resort to the following three constraints while designing a counterfactual input $z'$ for the LLM Machinery: i) $z'$ should be a counterfactual for $\mathcal{C}_1$, as our task is to measure how many counterfactuals for $\mathcal{C}_1$ are also counterfactual for $\mathcal{C}_2$. ii) $z'$ should not be representation of any data-point from $\mathcal{T}$, iii) It should be a transformation of the original data-point $x$ and its perturbation $\mu$.

We assume $z'$ has the following generic form (satisfying ii. and iii.), $z' = z + T(\mu)$, where $z = E(x)$ is an input to the LLM machinery. So, $z' = E(x) + T(\mu)$. Note that to ensure $T(\mu)$ is an invertible function of $\mu$ (satisfying iii.), we use an autoencoder which maps $\mu$ to $T(\mu)$ and then back to $\mu$ again. Finally, to satisfy the first constraint, we ensure the following holds true:

$$\mathcal{C}_1(E(x) + T(\mu)) = \mathcal{C}_1(E(x + \mu)) \tag{10}$$

Note that this can be enforced by standard KL divergence loss between $\mathcal{C}_1$ and $\mathcal{C}_2$.

---

**Algorithm 1:** Counterfactual F1 Score for $\mathcal{C}_1$ and $\mathcal{C}_2$

---

**Input:** Data-point $\boldsymbol{x} \in \mathcal{T}$
**Function** `CounterFactual(`$\boldsymbol{x}$`):`
    $\boldsymbol{x}' \leftarrow \arg\min_{\boldsymbol{x}' \in \mathcal{T}} d(\boldsymbol{x}, \boldsymbol{x}')$ s.t. $\mathcal{C}_1(\boldsymbol{x}) \neq \mathcal{C}_1(\boldsymbol{x}')$ ;
    $\mu \leftarrow \boldsymbol{x}' - \boldsymbol{x}$ ;               // Compute the perturbation
    $\boldsymbol{z} \leftarrow E(\boldsymbol{x})$ ;                 // Encode the original input
    $T(\mu) \leftarrow f(\mu)$ where $g(f(\mu)) = \mu$ ;     // Transform the perturbation
    $\boldsymbol{z}' \leftarrow \boldsymbol{z} + T(\mu)$ ;
    **return** $\boldsymbol{z}'$, $\boldsymbol{x}'$

**Procedure** *Calculate Counterfactual F1 score*
    ZList $\leftarrow$ [] ;
    XList $\leftarrow$ [] ;
    **while** $\mathcal{T} \neq \phi$ **do**
        Sample $\boldsymbol{x} \in \mathcal{T}$ ;              // Draw a new data point
        $\boldsymbol{z}', \boldsymbol{x}' \leftarrow$ `CounterFactual(`$\boldsymbol{x}$`)` ;
        **Ensure:** $\mathcal{C}_1(\boldsymbol{z}') = \mathcal{C}_1(E(\boldsymbol{x}'))$ ;        // constraint i.
        ZList $\leftarrow$ ZList $\cup \{\mathcal{C}_2(\boldsymbol{z}')\}$ ;     // Append $\mathcal{C}_2(\boldsymbol{z}')$ to the list
        XList $\leftarrow$ XList $\cup \{\mathcal{C}_1(\boldsymbol{x}')\}$ ;     // Append $\mathcal{C}_1(\boldsymbol{x}')$ to the list
        $\mathcal{T} \leftarrow \mathcal{T} - \{\boldsymbol{x}\}$ ;
    **return** $F_1 - score(\text{XList}, \text{ZList})$

---

Table 1: Ablation studies. $\mathcal{L}_{MSE}$ refers to an MSE loss between $c$ and $x$, such that $F(E(x)) = E(x)$. B-1, B-2, B-3, B- refers to Bleu scores with various $n$ gram precisions.

|  |  | F1 | B-1 | B-2 | B-3 | B-4 | BertScore |
|---|---|---|---|---|---|---|---|
| **Hateful Meme** | $\mathcal{L}_{\phi_2}$ | 97.29 | **0.65** | 0.53 | **0.47** | **0.39** | **0.971** |
|  | $\mathcal{L}_{\phi_1} + \mathcal{L}_{\phi_2}$ | 98.44 | **0.65** | 0.53 | **0.47** | **0.39** | **0.971** |
|  | $\mathcal{L}_{\phi_1} + \mathcal{L}_{\phi_2} + \mathcal{L}_{MSE}$ | **98.55** | 0.64 | **0.53** | 0.46 | 0.39 | 0.971 |
|  | $\mathcal{L}_{\phi_1} + \mathcal{L}_{\phi_2} + \mathcal{L}_C$ | 98.33 | 0.64 | 0.53 | 0.46 | 0.38 | 0.971 |
|  | $\mathcal{L}_{CAuSE}$ | 98.09 | 0.64 | 0.51 | 0.44 | 0.36 | 0.969 |
| **e-SNLI-VE** | $\mathcal{L}_{\phi_2}$ | 94.66 | 0.39 | 0.27 | 0.19 | 0.15 | 0.905 |
|  | $\mathcal{L}_{\phi_1} + \mathcal{L}_{\phi_2}$ | 94.08 | 0.39 | 0.27 | 0.19 | 0.15 | 0.905 |
|  | $\mathcal{L}_{\phi_1} + \mathcal{L}_{\phi_2} + \mathcal{L}_{MSE}$ | 94.39 | **0.39** | **0.27** | **0.20** | **0.15** | **0.905** |
|  | $\mathcal{L}_{\phi_1} + \mathcal{L}_{\phi_2} + \mathcal{L}_C$ | **94.94** | 0.38 | 0.27 | 0.20 | 0.15 | 0.905 |
|  | $\mathcal{L}_{CAuSE}$ | 91.96 | 0.39 | 0.27 | 0.20 | 0.15 | 0.904 |

Table 2: In addition to the Counterfactual F1 score, we also report the number of comprehensible generations (*#gen*), as many outputs from CAuSE tend to be gibberish when counterfactual input is provided. To provide a more holistic evaluation of CAuSE's performance on counterfactual inputs, we compute the harmonic mean (HM) of the F1 score and *#gen*, capturing both accuracy and the quality of generated explanations.

|  | **Hateful Meme** | | | **e-SNLI-VE** | | |
|---|---|---|---|---|---|---|
|  | F1 | # gen. | HM | F1 | # gen. | HM |
| $\mathcal{L}_{\phi_1} + \mathcal{L}_{\phi_2}$ | 55.02 | 17 | 32.98 | **93.81** | 167 | 28.35 |
| $\mathcal{L}_{\phi_1} + \mathcal{L}_{\phi_2} + \mathcal{L}_{MSE}$ | 33.33 | 2 | 3.976 | 85.94 | 322 | 46.84 |
| $\mathcal{L}_{\phi_1} + \mathcal{L}_{\phi_2} + \mathcal{L}_C$ | 53.78 | 91 | 15.56 | 73.48 | 850 | 78.82 |
| $\mathcal{L}_{CAuSE}$ | **75.81** | **755** | **75.61** | 85.24 | **986** | **91.43** |

## 4 RESULTS AND ANALYSIS

### 4.1 AUTOMATIC EVALUATION

The proposed system is evaluated across two verticals: i) Mimicking capability of the *explanator* when compared to *post-hoc classifier*, and ii) performance under counterfactual input. The automatic evaluation metric used to evaluate CAuSE performance can be grouped into two categories, i) **Faithfulness:** This is measured by the obtained F1 score measured between the predicted class by the LLM machinery (or $\mathcal{C}_2$) and the predicted class by the post-hoc classifier $\mathcal{C}_1$. The predicted class obtained from the LLM machinery is extracted either from the prediction of $\phi_2$ or from $\mathcal{C}_2$ classifier head. ii) **Plausibility:** This is measured as the BLEU scorePapineni et al. (2002) and BERTScoreZhang et al. (2020) between the generated explanation and the ground truth explanation from the test set.

**Baselines.** *To the best of our knowledge, ours is the first approach that generates **faithful** natural language explanations directly from a classifier's hidden state.* Nonetheless, we compare our method with several Visual Language Model (VLM) baselines as there are no existing techniques for this task in the literature. Specifically, we use zero-shot and few-shot ($k = 2$ or 3) prompting with i) PaLiGemma(Beyer et al., 2024), ii) LLaVA(Liu et al., 2023), to simulate the predicted class from a given classifier ($\mathcal{C}_1$), based on previous input-output examples[2]. Since it is challenging to simulate a model's behaviour without access to its hidden activations, few-shot prompting often performs similarly or even worse than zero-shot prompting. The faithfulness of the explanations, as measured by the F1 score, is inconsistent and random (below 50% for the Hateful Memes dataset and below 33% for e-SNLI-VE), as shown in Table 3 The fine-tuned models (shown through FT suffix) perform the best, where the F1 score reaches close to $\sim 70\%$.

---

[2]The specific prompting used are shown in the Appendix C

Table 3: Various VLM-based baselines. FT as a suffix denotes finetuned model. Note that LLaVA has 7B and PaLiGemma has 3.5B parameters respectively.

| Dataset | Baselines | F1 | B-1 | B-2 | B-3 | B-4 | BertScore |
|---------|-----------|-----|-----|-----|-----|-----|-----------|
| **Hateful Meme** | *LLaVA-0-shot* | 58.44 | 0.09 | 0.01 | 0.01 | 0.01 | 0.889 |
| | *LLaVA-2-shot* | 46.55 | 0.12 | 0.02 | 0.01 | 0.01 | 0.864 |
| | *PaLiGemma -FT* | 72.33 | 0.41 | 0.27 | 0.15 | 0.09 | 0.891 |
| | *LLaVA-FT* | 72.38 | 0.40 | 0.27 | 0.17 | 0.13 | 0.894 |
| **e-SNLI-VE** | *LLaVA-0-shot* | 33.12 | 0.22 | 0.07 | 0.03 | 0.02 | 0.876 |
| | *LLaVA-3-shot* | 35.77 | 0.22 | 0.07 | 0.03 | 0.01 | 0.869 |
| | *PaLiGemma -FT* | 64.90 | 0.19 | 0.04 | 0.01 | 0.01 | 0.866 |
| | *LLaVA-FT* | 64.29 | 0.22 | 0.08 | 0.03 | 0.02 | 0.859 |

Table 4: Case studies: A few example where our model succeeds. **Pred:** Explanation generated from the model, **GT:** Ground truth explanation.

| Image Path | Pred | GT | $y_1$ | $y_2$ |
|------------|------|-----|-----|-----|
| 489134459.jpg | A woman is a female. Just because she is sitting on a curb, it means she is outside.. | A boy and a girl are two kids. The front of a house is located outside.. | E | E |
| 5631556013.jpg | A man is performing on the street in front of a group of people.. | man jumping from someon | E | E |
| 12507.png | it promotes negative stereotypes about people who are Muslim and suggests that all Muslims are violent or dangerous | it promotes harmful stereotypes about Muslims, suggesting that they are violent and intolerant. | O | O |
| 91462.png | it promotes racism, specifically by implying that white people are superior to other people. | it promotes harmful stereotypes about black women. | O | O |

### 4.1.1 ABLATION STUDIES

**What is the use of various loss function other than $\mathcal{L}_{\phi_1}$ and $\mathcal{L}_{\phi_2}$ loss?** As seen from Table 1, it can be posed as a valid question. Indeed, when using our proposed method which uses $\mathcal{L}_{IIT}$ and other losses seem to achieve slightly lower F1 score (indicating slightly lower faithfulness) and slightly lower BLEU score / BERTScore (indicating slightly lower plausibility). Note that this difference is very small and it is compensated by very high counterfactual F1 score as shown in Table 2 obtained by our method.

**Why is IIT required?** As can be seen from Table 1 and Table 2, good mimicking performance under normal condition does not always entail good performance when posed with counterfactual input. IIT ensures causal abstraction between $\mathcal{C}_2$ and $\mathcal{C}_1$ theoretically and this is also being verified empirically by the high counterfactual F1 score obtained by our method which uses IIT.

**Is $\mathcal{L}_{\phi_1}$ necessary?** $\mathcal{L}_{\phi_1}$ which is used to train the LLM ($\phi_1$) which reconstructs the content is required, because that shows better mimicking performance (at least in Hateful meme dataset) coupled with $\mathcal{L}_{\phi_2}$ than using $\mathcal{L}_{\phi_2}$ alone. This can be attributed to the joint training objective which ensures that $c$ possesses enough input information to aid in explanation generation by the second LLM $\phi_2$.

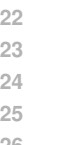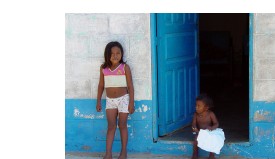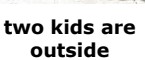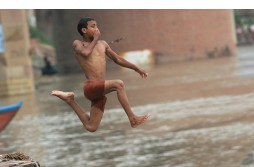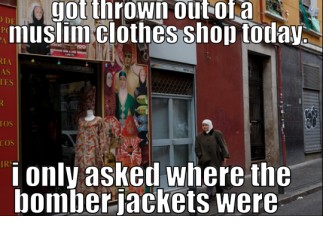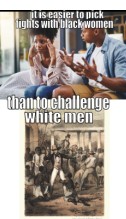

**two kids are outside**          **There is a man jumping from someone**          i only asked where the bomber jackets were

Figure 3: Examples corresponding to Table 4

Table 5: Error Analysis: These cases demonstrate four kinds of error cases that is prevalent among our proposed framework CAuSE.

| Image Path | Gen | GT | $y$ | $\hat{y}$ |
|---|---|---|---|---|
| 7046014201.jpg | Construction work necessitates working outdoor. | A juggler is juggling clubs at an outdoor plaza. | E | E |
| 2731298834.jpg | A dog that is jumping into the water will be wet. | swimming is perform in a water. | E | E |
| 151215569.jpg | A man is pulling on the street so he is outdoors. | A young blond girl describes a child and a man describes and adult. | E | E |
| 59260.png | it suggests that white people are superior to other people, which is not accurate | it promotes anti-Semitism and hatred towards Jewish people. | O | O |

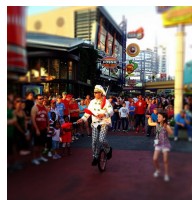 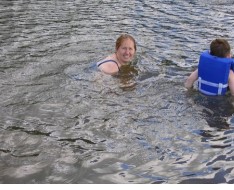 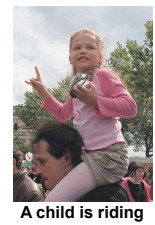 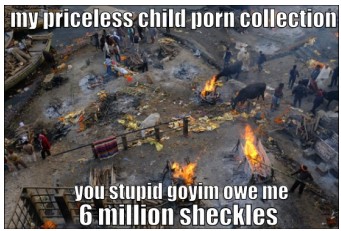

A juggler is performing outdoors.    There is water    A child is riding an adult.

Figure 4: Memes pertaining to Error Analysis shown in Table 5

## 4.2 QUALITATIVE STUDIES

### 4.2.1 CASE STUDIES

In Table 4, we present four successful examples from the e-SNLI-VE and Facebook Hateful Memes datasets (two from each). The first two examples are from e-SNLI-VE, while the latter two are from the Hateful Memes dataset. In the e-SNLI-VE examples, CAuSE produces semantically accurate explanations while correctly predicting the class as "Entailment." A noticeable pattern emerges from these successful cases: CAuSE tends to perform well when the class-level information can be explicitly inferred from the combination of the image and text. Specifically, for e-SNLI-VE, when CAuSE generates accurate explanations, the hypothesis often functions like a caption for the image premise, which aids in classification.

For the Hateful Memes examples, CAuSE also generates correct explanations. In these cases, the image and the embedded text are semantically aligned rather than contradictory (i.e., where the image-text mismatch is used to evoke negative sentiment). In such instances, CAuSE effectively provides explanations and correctly predicts the appropriate output class.

### 4.2.2 ERROR ANALYSIS

We selected four examples from the e-SNLI-VE and Hateful Memes datasets to highlight common types of errors made by CAuSE (in Table 5). These errors can be categorized into three main types:

**Lack of representation capability:** In the first example, the hypothesis reads, "A juggler is performing outdoors," and the premise is entailed, as confirmed by the ground truth explanation: "A juggler is juggling clubs at an outdoor plaza." However, CAuSE incorrectly generates the explanation: "Construction work necessitates working outdoors," confusing the act of juggling with construction work. This error likely stems from insufficient information in the initial representation, $c$, used by CAuSE.

**Lack of object-level representation:** The post-hoc classifier relies on unimodal representations from the CLIP architecture, which lacks fine-grained object-level details, compared to models like Faster R-CNNRen et al. (2016). In the second example, instead of recognizing a "dog," CAuSE should have identified "a woman and children" for a more accurate representation.

The third example illustrates both issues: lack of object-level representation and general representation capability. These limitations prevent CAuSE from correctly describing the relationship between "a young blonde girl," "an adult," and "a man pulling outdoors."

**Implicit semantic category:** In the fourth example, although CAuSE correctly predicts the output class as offensive, it does so for the wrong reasons. Even a human might struggle to recognize the

implicit anti-Semitism in this meme, as neither the image nor the text explicitly convey the historical context of the Holocaust, where six million Jews were killed. Without this prior knowledge, CAuSE cannot fully comprehend the offence.

## 5 RELATED WORK

**Interpretabiltiy**. Interpretability is crucial for building trust in AI systems within human society. Techniques like LIME, SHAP and RISE (Ribeiro et al., 2016; Lundberg & Lee, 2017; Petsiuk et al., 2018) explain classifier predictions by providing feature-level explanations for local interpretability. Although model-agnostic, these methods lack global interpretability, which is addressed by GALE van der Linden et al. (2019), where local explanations are aggregated into a global model understanding. Approaches like SmoothGrad Smilkov et al. (2017) and Integrated Gradients (Sundararajan et al., 2017b) utilize input gradients for model explanation, while CAM Zhou et al. (2015) highlights critical pixels for decision making in visual classification. Counterfactual generations (Chang et al., 2019; Mothilal et al., 2020; Goyal et al., 2019) also offer insights into the inner working of the model by revealing decision boundaries. However, most of these methods often overlook implicit features behind model decisions and lack natural language explanations. To address these limitations, we propose a novel framework for classifier explanations which generates both *faithful* and *plausible* natural language outputs.

**Causal Interpretability**. Causal interpretability refers to the ability to explain a model's decisions by identifying the cause-effect relationships between input features and the model's output. Feder et al. (2022) demonstrated how incorporating causal reasoning in NLP tasks can improve model predictions and enhance interpretability by going beyond simple correlations between input features and outputs. Further works by Geiger et al. (2021b); Vig et al. (2020); Meng et al. (2023) have focused on causal abstraction and causal mediation analysis, helping to create causally faithful models and identify both direct and indirect causal factors behind certain model behaviors. In addition to generating counterfactuals, testing models on counterfactual inputs is another critical aspect of understanding model behavior. Since creating exact counterfactuals is challenging, Abraham et al. (2022); Calderon et al. (2022), recent research has focused on approximations Geiger et al. (2021b) or counterfactual representations Feder et al. (2021); Elazar et al. (2021); Ravfogel et al. (2021). Our proposed counterfactual metric is inspired by these counterfactual representations. Moreover, most of the existing works focuses on single modality (e.g., text or vision) Feder et al. (2021); Goyal et al. (2020). In contrast, the natural language causal explanation provided by our framework is model-agnostic, task-agnostic, and capable of handling multimodal inputs.

## 6 CONCLUSION AND FUTURE WORK

In this paper, we presented *CAuSE* (Causal Abstraction under Simulated Explanation), a novel framework for generating causally faithful natural language explanations for multimodal classifiers. By integrating *Interchange Intervention Training* (IIT) with a Language Model (LM) based module, CAuSE addresses the limitations of existing interpretability methods, ensuring explanations are directly tied to the classifier's causal reasoning. Our new Counterfactual F1 score highlights CAuSE's state-of-the-art performance on datasets like e-SNLI-VE and Facebook Hateful Memes.

While CAuSE demonstrates robust task-agnostic performance, future work will focus on enhancing fine-grained object-level representations and extending the framework to temporal data, such as video and audio. Additionally, we aim to explore how *self-supervised* learning and deeper integration of implicit cultural knowledge can further improve the framework's scalability and contextual understanding in real-world applications.

## ETHICS STATEMENT

The datasets used in this study are publicly available. The explanations for hateful memes were generated from publicly accessible meme data, and we adhered to copyright regulations to prevent any infringement. Furthermore, our research received approval from the Institutional Review Board (IRB). Since the hateful meme dataset includes content that may be offensive, we recommend that readers approach it with discretion.

## REPRODUCIBILITY STATEMENT

To ensure reproducibility, we consistently use a random seed of 42 across all experiments. The code is available at `https://anonymous.4open.science/r/CAuSE-5BD0`, and model outputs will be shared upon paper acceptance. These outputs can be cross-verified with the results generated from the provided code. Our method is theoretically sound, supported by the proof of the proposed theorem and proposition outlined in Appendix A, with all underlying assumptions clearly stated and justified.

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

# A THEOREMS

During our training process, we have implemented IIT (Interchange Intervention Training) along with an additional constraint that the weights of $C_1$ and $C_2$ become the same during training, which is ensured by the frobenius norm term being used as a part of $\mathcal{L}_{CAuSE}$

**Theorem 1.** Under the above stated conditions, when $C_2$ becomes a causal abstraction of $C_1$ and their weights become the same, $E(x) = F(E(x))$.

*Proof.* Without loss of generality, we have considered only two input neurons for both $C_1$ and $C_2$. Under IIT, the following always holds between two output neurons (we assume that a) there exists one intermediate intervened neuron and b) source $s$ and base $b$ inputs are provided as input):

$$softmax(y_1) = softmax(y_2)$$

$$\implies softmax(w_{1i}(E(s)_i), w_{2i}(E(b)_i)) = softmax(w'_{1i}F(E(s)_i), w'_{2i}F(E(b)_i)), \forall i = 1, 2$$

$$\text{Since } w = w'$$

$$\implies \frac{exp(\sum_i w_{1i}E(b)_i}{exp(\sum_i w_{1i}E(b)_i + exp(\sum_i w_{2i}E(s)_i} = \frac{exp(\sum_i w_{1i}F(E(b))_i}{exp(\sum_i w_{1i}F(E(b))_i + exp(\sum_i w_{2i}F(E(s))_i}$$

$$\forall i = 1, 2 \tag{11}$$

Let us assume $E(b) \neq F(E(b))$ so there $\exists$ j such that $E(b)_j \neq F(E(b))_j$. Let us assume,

$$E[b] = [p, q]$$

$$F[E[b]] = [p, \rho q] \tag{12}$$

We pick $s, b \in \mathcal{D}_E \times \mathcal{D}_E$ so $\exists s, b$ such that $s = b$ when $s_i = b_i$. Here $\mathcal{D}_E$ refers to the data on which the encoder is being trained. Finally, from Equation 11,

$$
\begin{aligned}
w_{21}[E(s)_2 - F(E(s))_2] = &\log\left(\exp(w_{12}E(b)_1 + w_{22}E(b)_2) + \exp(w_{11}E(s)_1 + w_{21}E(s)_2)\right) \\
&- \log(\exp(w_{12}F(E(b))_1 + w_{22}F(E(b))_2) + \exp(w_{11}F(E(s))_1 \\
&+ w_{21}F(E(s))_2))
\end{aligned}
\tag{13}
$$

Here, we name all the variables to for better readability:

$$
w_{11} = \beta \quad w_{12} = \gamma \quad w_{21} = \delta \quad w_{22} = \epsilon
\tag{14}
$$

From these, considering $s = b$ we can rewrite the equation 13 for two output neurons as:

$$
\delta q(1 - \rho) = \underbrace{\log(\exp(\beta p + \delta q) + \exp(\gamma p + \epsilon q))}_{\mathcal{P}_1} - \underbrace{\log(\exp(\beta p + \delta \rho q) + \exp(\gamma p + \epsilon \rho q))}_{\mathcal{P}_2}
\tag{15}
$$

$$
\epsilon q(1 - \rho) = \mathcal{P}_1 - \mathcal{P}_2
\tag{16}
$$

This means, under IIT if :

$$
\delta q(1 - \rho) = \mathcal{P}_1 - \mathcal{P}_2 + k_1
\tag{17}
$$

and

$$
\epsilon q(1 - \rho) = \mathcal{P}_1 - \mathcal{P}_2 + k_2
\tag{18}
$$

then , $k_1 = k_2 = 0$

Now, we impose pairwise, equality of weights $\beta = \gamma$ and $\delta = \epsilon$. Using this condition, the individual equations will become:

$$
\epsilon q(1 - \rho) = \epsilon q(1 - \rho) + k_1
\tag{19}
$$
$$
\epsilon q(1 - \rho) = \epsilon q(1 - \rho) + k_2
\tag{20}
$$
$$
\implies k_1 = k_2 = 0
\tag{21}
$$

The above condition where we consider pairwise equality of weights is a degenerate case. In this situation, every input node has the same weightage as it is passed to the deeper layers. This is an edge case rarely seen in real training scenarios.

We obtain the following values of k when the degenerate case is not considered:

$$
\begin{aligned}
k_1 = &\log\left(\exp(\beta p + \delta \rho q) + \exp(\gamma p + \epsilon \rho q)\right) \\
&- \log\left(\exp(\beta p + \delta q) + \exp(\gamma p + \epsilon q)\right) \\
&+ \delta q(1 - \rho)
\end{aligned}
\tag{22}
$$

$$
\begin{aligned}
k_2 = &\log\left(\exp(\beta p + \delta \rho q) + \exp(\gamma p + \epsilon \rho q)\right) \\
&- \log\left(\exp(\beta p + \delta q) + \exp(\gamma p + \epsilon q)\right) \\
&+ \epsilon q(1 - \rho)
\end{aligned}
\tag{23}
$$

The above values of $k_i \neq 0 \quad \forall i = 1, 2$. However, this is a contradiction since, this violates the property of (IIT). This means our initial assumption of $F(E(b)) \neq E(b)$ is wrong. This proves that $F(E(b)) = E(b)$.

Also, it is noteworthy that in the equations 19 $\quad k_1 = k_2 = 0$ only when $\rho = 1$. This again validates our claim that $E(b) = [p, q] = F(E(b))$ $\qquad \square$

*Definition.* LLM machinery $F$ coupled with the classifier $\mathcal{C}_2$ is called *Explanator*, while the encoder with the classifier $\mathcal{C}_1$ is called the *Post-hoc classifier*. We also assume the following, there exists a function $\delta$ which maps a set of variables $(V_E)$ in $E$ to a set of variables $(V_F)$ in $F$, such that $\delta : V_E \to V_F$. We also assume that we intervene in a neuron $i_e \in V_E$, such that a mapped neuron $\delta(i_e) \in V_F$ is also intervened. Under this intervention schema, the intervened outputs of $E$ and $F$ are denoted as $F^{INT}(E(s), E(b))$ and $E^{INT}(s, b)$. The following lemma shows their relation.

*Lemma 1.* If $\mathcal{C}_1$ and $\mathcal{C}_2$ become identical (their weights are equal and they are causal abstraction of each other), $F^{INT}(E(s), E(b)) = E^{INT}(s, b)$, meaning $F$ is a causal abstraction of $E$.

*Proof.* By Theorem 1, we know that if $\mathcal{C}_1$ and $\mathcal{C}_2$ become identical, then $F(E(x)) = E(x)$. This entails $F$ acts as a perfect autoencoder considering $F$ only accepts input from $E$. The source $(s)$ and base $(b)$ equivalent input for $F$ would be $E(s)$ and $E(b)$, respectively. When supplied with $E(s)$ and $E(b)$ and an interchange intervention is performed in $F$, the $F$ being a perfect autoencoder will try to reconstruct $E(s)$ but due to intervention with $E(b)$, the output will also contain a part of $E(b)$. $F$ being a linear function of $E(x)$, we can write:

$$F^{INT}(E(s), E(b)) = f_1(E(s), E(b), w_F)E(s) + f_2(E(s), E(b), w_F)E(b) \tag{24}$$
$$\psi_1 E(s) + \psi_2 E(b) \tag{25}$$
$$[\text{as} \quad F^{INT}(x) = F(x) = x \quad \text{and by previous argument}] \tag{26}$$

Note that $E^{INT}(s, b) = E(s, b)$ (denoting a function of $(s, b)$) and as $E$ is a linear function of $s, b$, by similar argument:

$$E^{INT}(s, b) = E(s, b) = g_1(s, b, w_E)E^{INT}(s) + g_2(s, b, w_E)E^{INT}(b) \tag{27}$$
$$= \phi_1 E(s) + \phi_2 E(b) \quad [\text{as} \quad E^{INT}(x) = E(x)] \tag{28}$$

If the contribution of $E(s)$ towards its reconstruction by $F$, as quantified by $\psi_1$ and its equivalent contribution (through $s$) towards the intervention in $E$ are the same then $\psi_1 = \phi_1$. Physically this means $s$ is as important to $E$ as $E(s)$ is important to $F$ for any equivalent $b$ and $E(b)$. This is satisfied trivially when $E$ and $F$ give equal importance to any data point $x$ and its transformed version $E(x)$, which is exactly ensured in training by the fact that $E(x) = F(E(x))$. This further implies $\frac{\psi_1}{\phi_1} \to 1$, and $\frac{\psi_2}{\phi_2} \to 1$. We observe, $F^{INT}(E(s), E(b)) = E^{INT}(s, b)$.

$\square$

**Theorem 2:** If $\mathcal{C}_1$ and $\mathcal{C}_2$ become identical (their weights are equal and they are causal abstraction of each other), the *Explanator* becomes a causal abstraction of the *Post-hoc classifier*.

*Proof.* From lemma 1, we showed $F$ is a causal abstraction of $E$. For any intervention performed between $E$ and $F$, their outputs are equal, which are being fed to $\mathcal{C}_1$ and $\mathcal{C}_2$ respectively. As for the same input, $\mathcal{C}_1$ and $\mathcal{C}_2$ outputs will match, the final output from the explanator and post hoc classifier will also match. If any intervention is performed between $\mathcal{C}_1$ and $\mathcal{C}_2$, their output will also match because they were trained to be causal abstraction of each other.

So in summary, for pairwise interchange intervention between $E$ and $F$ or $\mathcal{C}_1$ and $\mathcal{C}_2$, the final output from post-hoc classifier and explanator will match. This is the definition for causal abstraction. Therefore, the explanator becomes a causal abstraction of the post-hoc classifier. $\square$

## B  SOME GENERIC THEORETICAL RESULTS

**Notation:** Assume two identical neural nets $N_1$ and $N_2$. Their weights are $w$ and $\hat{w}$. These two neural nets are trained on two different datasets: $\mathcal{D}_1$ and $\mathcal{D}_2$. We denote activation at an arbitrary layer's neuron as $i_n$, where the subscript $i$ denotes the NNs. We also assume $k = \frac{i_2}{i_1}$. The following lemma shows a relation.

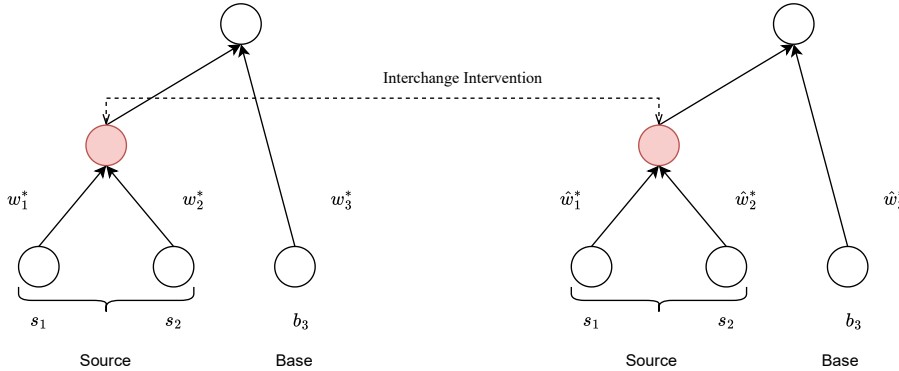

Figure 5: Structure of $N_1$ and $N_2$ are assumed to be the same. The red neuron denotes the intervened neuron.

*Lemma 2.* For $N_1$ and $N_2$, after convergence, $k = f(\mathcal{D}_1, \mathcal{D}_2, x)$, where $x$ denotes the network input.

*Proof.* After training is complete, assume the optimal weights are $w_1^*$ and $w_2^*$. Naturally, $w_1^* = \psi_1(\mathcal{D}_1)$ and $w_2^* = \psi_2(\mathcal{D}_2)$. $k$, being a ratio of the activation of two neural nets, will depend on their inputs, and converged weights. Therefore, $k = f(w_2^*, w_1^*, x) = f(\mathcal{D}_1, \mathcal{D}_2, x)$. □

*Lemma 3.* If outputs under interchange intervention are equal (as satisfied by IIT training objective), then $k$ must be of the form $f(\mathcal{D}_{IIT}, s, b)$ to ensure $N_2$ and $N_1$ are the causal abstraction of each other.

*Proof.* Suppose, the neural networks converge to a state where both $N_1$ and $N_2$ have the parameter values $w^*$ and $\hat{w}^*$. Refer to Figure 5.

If the outputs are equal after intervening on $V_1$, and identically $V_2$ neuron of $N_1$ and $N_2$ respectively, then:

$$\underbrace{w_1^* s_1 + w_2^* s_2}_{i_1} + w_3^* b_3 = \underbrace{\hat{w}_1^* s_1 + \hat{w}_2^* s_2}_{k i_1} + \hat{w}_3^* b_3 \tag{29}$$

Assume $k = f(\mathcal{D}_{IIT}, s, b, \alpha)$. From Equation 29,

$$f(\mathcal{D}_{IIT}, s, b, \alpha) = 1 - \frac{\psi(w^*, \hat{w}^*, s, b)}{i_1} \tag{30}$$

We know IIT optimizes the weights such that two neural nets become causal abstraction of each other. This is done essentially by confounding on $\mathcal{D}_{IIT}$, where $s, b \sim \mathcal{D}_{IIT}$ and $\mathcal{D}_{IIT} \to w^*$ and $\mathcal{D}_{IIT} \to \hat{w}^*$.[3] This argument necessistates the RHS of Equation 30 depends only on $\mathcal{D}_{IIT}$, $s$, and $b$. For LHS to be equal, it also must depend on these parameters, facilitating $\alpha$ as a spurious variable.

The equation itself is the necessary as well as the sufficient condition (i.e. the definition) for causal abstraction which is satisfied when $k = f(\mathcal{D}_{IIT}, s, b)$. Thus i) Equality of output of two neural nets under interchange intervention and ii) $k = f(\mathcal{D}_{IIT}, s, b)$ together pose as *a necessary and sufficient condition* for causal abstraction between $N_1$ and $N_2$

□

*Note:* Although this is shown for the above neural network having specific architecture, this holds true regardless of the architecture, as the functional form of RHS and LHS must match.

---

[3] $\to$ denotes the causal arrow, i.e. by optimizing on $\mathcal{D}_{IIT}$, we obtain both $w^*$ and $\hat{w}^*$

We assume $E$ and $F$ are encoder and LLM machinery respectively having weights of $w_E$ and $w_F$. The encoder and LLM machinery are followed by $\mathcal{C}_1$ and $\mathcal{C}_2$ respectively having weights $w$ and $\hat{w}$. Let us assume $w_E^*$ is the optimized weight of the encoder when it is fine-tuned with $x \sim D_E$. Further, assume we have done IIT on $\mathcal{C}_1$ and $\mathcal{C}_2$ keeping the encoder frozen.

Following would be the dependency of various weights: i) $w_E^* = f_1(\mathcal{D}_E)$, ii) $w_F^* = f_2(w_E^*, \mathcal{D}_{IIT}, w^*, \hat{w}^*)$, iii) $w^* = f_3(\mathcal{D}_{IIT})$ and iv) $\hat{w}^* = f_4(\mathcal{D}_{IIT})$. Without loss of generality we can assume $\mathcal{D}_{IIT} = \mathcal{D}_E$, as both $s, b$ and $x$ are being sampled from the same dataset. The functional dependencies then boil down to the fact that all the weights are a function of $\mathcal{D}_{IIT}$.

Being a closely trained system with only one dataset $\mathcal{D}_E = \mathcal{D}_{IIT}$, and from lemma 1, the most generalized version linking the intervened output from $F$ and $E$ will be $F^{INT}(E(s,b)) = \phi(\mathcal{D}_{IIT}, s, b)E^{INT}(s,b)$.

Upon the assumption that $E$ and $F$ are two variables (i.e. neurons, composed of all other neurons inside $E$ and $F$) inside $\mathcal{C}_1$ and $\mathcal{C}_2$ respectively, their intervened output depends only on $\phi(\mathcal{D}_{IIT}, s, b)$. Also, $E$ and $F$ are assumed to be be inside $\mathcal{C}_1$ and $\mathcal{C}_2$ respectively would mean for any input $(s, b)$, their intervened outputs remain the same. Both of these satisfy the necessary and sufficient requirements for causal abstraction as per Lemma 2. This is complementary to Theorem 2 and its proof, shown by assuming these strong conditions.

## C  PROMPTS

---

**0-shot Meme Dataset Prompt**

**Prompt:** Is this image offensive?  If it is offensive, give a single-line explanation, otherwise simply state that it is 'not offensive'.

images/sample_image0.png

---

**Few-shot Meme Dataset Prompt**

**Prompt 1:** Is this meme offensive?  Answer briefly.  Give 1 line explanation only if it is offensive.

images/sample_image1.png

**Assistant:** This meme is offensive. {Explanation} goes here.

**Prompt 2:** Is this image offensive?  Answer briefly.  Give 1 line explanation only if it is offensive.

images/sample_image2.png

**Assistant:** This meme is not offensive.

**Prompt 3:** Is this image offensive?  Answer briefly.  Give 1 line explanation only if it is offensive.

images/sample_image3.png

---

**0-shot SNLI VE Prompt**

**Prompt:** Answer with 'entailment', 'contradiction', or 'neutral' if the hypothesis that *[Insert hypothesis here]* follows the image, contradicts it, or is neutral to it. Also, give a 1-line explanation for your answer.

images/sample_image_snli.png

---

**Few-shot SNLI VE Prompt**

**Prompt 1:** Answer with 'entailment', 'contradiction', or 'neutral' if the hypothesis that *[Insert **entailment hypothesis**]* follows the image, contradicts it, or is neutral to it. Also, give a 1-line explanation for your answer.

images/sample_image_snli_0.png

**Assistant:** Entailment. {Explanation} goes here.

**Prompt 2:** Answer with 'entailment', 'contradiction', or 'neutral' if the hypothesis that *[Insert **contradiction hypothesis**]* follows the image, contradicts it, or is neutral to it. Also, give a 1-line explanation for your answer.

images/sample_image_snli_1.png

**Assistant:** Contradiction. {Explanation} goes here.

**Prompt 3:** Answer with 'entailment', 'contradiction', or 'neutral' if the hypothesis that *[Insert **neutral** hypothesis]* follows the image, contradicts it, or is neutral to it. Also, give a 1-line explanation for your answer.

images/sample_image_snli_2.png

**Assistant:** Neutral. {Explanation} goes here.

## D    DATASET AND EXPERIMENTATION

The e-SNLI-VE dataset includes human-annotated explanations for both text and images. For offensive memes, we generated explanations using the Jurassic-1[4] language model through zero-shot prompting, as detailed in Appendix Section C via its relevant prompts. In this context, the LLM-generated explanations serve as the ground truth. The experiments were conducted on a Kaggle kernel with PyTorch version 2.1.2 and a single P-100 GPU, with a random seed of 42 maintained for all runs. Additionally, baseline VLM models were implemented using PEFT[5] and LoRA(Hu et al., 2021). The code is available anonymously for review.

Table 6: Train-test splits for e-SNLI-VE and Hateful Memes datasets.

| Dataset | Train Split | Test Split |
|---|---|---|
| e-SNLI-VE | 9000 | 1000 |
| Hateful Memes | 6997 | 1000 |

---

[4]https://www.ai21.com/blog/announcing-ai21-studio-and-jurassic-1
[5]https://github.com/huggingface/peft

