# OpenReview forum: "CAuSE: Post-hoc Natural Language Explanation of Multimodal Classifiers through Causal Abstraction"
_ICLR.cc/2025/Conference — Submitted to ICLR 2025_

### Official Review · Reviewer_s4tx · 2024-10-31

**Soundness:** 1
**Presentation:** 2
**Contribution:** 2
**Rating:** 3
**Confidence:** 3

**Summary:**

This paper proposes a new framework for natural language explanations (NLEs) of Vision Language Models (VLMs). It introduces a new metric, counterfactual F1 score with which to self-assess its method.

**Strengths:**

- The paper makes strides on a relevant topic (NLEs for VLMs). Much work appears to have gone into architectural/training design.
- The subject area is one of importance. Many results are amassed.

**Weaknesses:**

- While the paper seems to make good theoretical strides, it is nearly impossible to evaluate the given method against relevant baselines. The paper does verify that this is the first method of its type to compute NLEs directly from a classifier's hidden state. However, it does lack comparison against any other methods (black box or not) that would help significantly.
- Readability is a major problem in the paper. Definitions and diagrams are piled somewhat haphazardly atop one another. Occasionally, vague justifications are used i.e. *Recent studies (Madsen et al., 2024) have highlighted these faithfulness issues, revealing inconsistencies when models are further probed* (not enough room is given for exposition).
- The claims are elaborate in the paper (such as state-of-the-art performance on the justification that no other methods exist, as above), but the experimental evaluation is too small to justify such claims. I would recommend toning these down to some degree: *this ensures that CAuSE’s natural language explanations are not only simulations of the classifier’s behavior but also reflect its underlying causal processes*- How is it ensured? The paper later shows failure cases, which makes this statement confusing and the claim to be slightly too strong.
- Typo: L49 *Visual Language Models*

**Questions:**

- Why exactly are we unable to compare the method against other baselines? Please elaborate.

---

> ### Author Response · Authors · 2024-11-19
> **Response to Weaknesses**
>
> > While the paper seems to make good theoretical strides, it is nearly impossible to evaluate the given method against relevant baselines. The paper does verify that this is the first method of its type to compute NLEs directly from a classifier's hidden state. However, it does lack comparison against any other methods (black box or not) that would help significantly.
>
> Most existing post-hoc interpretability methods focus on identifying implicit or explicit concepts behind model decisions rather than generating faithful natural language explanations (NLEs). While approaches like VLM prompting or NLX-GPT address model behavior explanation, they differ from our work as we specifically focus on post-hoc NLEs. Our CAuSE model uniquely generates "faithful;" NLEs by leveraging the hidden states of a multimodal pre-trained classifier encoder, making it comparable only to the VLM prompting-based strategies.
>
> > Readability is a major problem in the paper. Definitions and diagrams are piled somewhat haphazardly atop one another. Occasionally, vague justifications are used i.e. Recent studies (Madsen et al., 2024) have highlighted these faithfulness issues, revealing inconsistencies when models are further probed (not enough room is given for exposition).
>
> Thank you for your sincere comments. We will try to improve the presentation of the paper to make it more readable. As for line 52 indicating that LLMs are inconsistent during explanation generation, we refer to the paper (Madsen et. al.) where it is shown that LLMs confidently generate explanations for many tasks that they excel at, however, their explanations are not consistent under  counterfactual setting, showing that LLM generated explanations are not self-consistent.
>
> > The claims are elaborate in the paper (such as state-of-the-art performance on the justification that no other methods exist, as above), but the experimental evaluation is too small to justify such claims. I would recommend toning these down to some degree: this ensures that CAuSE’s natural language explanations are not only simulations of the classifier’s behavior but also reflect its underlying causal processes- How is it ensured? The paper later shows failure cases, which makes this statement confusing and the claim to be slightly too strong.
>
> We have implemented our model on 2 different multimodal classifiers for 2 different classification tasks involving Hateful Meme prediction and SNLI-VE datasets. In both cases, our proposed model achieved the best results (counterfactual F1 score in Table 2) when compared to other techniques leveraging the hidden state of the encoder of the multimodal classifier as shown in the ablation studies 4.1.1. The multimodal encoder that is used in the post-hoc classifier is a CLIP-based model involving late fusion.
>
> We have implemented the Interchange Intervention Training (IIT) paradigm in CAuSE. This ensures that the LLM machinery in the CAuSE framework generates causally faithful explanations of the pre-trained multimodal classifier. The simple F1 score between the multimodal classifier (post-hoc classifier) and our CAuSE framework is high, which indicates high agreement between the two of them under regular testing circumstances. We have termed this behaviour of CAuSE as a “simulation” of the post-hoc classifier. However, the counterfactual F1 score between CAuSE and the post-hoc classifier is also quite high when compared to other ablation techniques under counterfactual inputs. The ability of CAuSE to mimic the behaviour of the post-hoc classifier under altered counterfactual inputs is indicative of the causal abstraction between the two.
>
> Our model is limited by the bottleneck arising from the single representation of the hidden state of the multimodal encoder of the pre-trained classifier (post-hoc classifier) that is used in CAuSE. We have mentioned this in error analysis 4.2.2 line 475. Furthermore, our model’s LLM machinery could be implemented with any state-of-the-art LLMs or VLMs that might improve its generation quality. However, these examples are few and do not represent the entire performance of our model.

---

> ### Author Response · Authors · 2024-11-19
> **Response to question**
>
> > Why exactly are we unable to compare the method against other baselines? Please elaborate.
>
> Most post-hoc interpretability techniques revolve around finding implicit (e.g. Integrated gradient [1]) and explicit concepts (e.g. Semantify [2]) that drive the model decision instead of generating coherent natural language explanations (NLEs). The task of “faithful” natural language explanation of model behaviour has not been seriously tackled before. A naive method of post-hoc model behaviour explanation using NLEs prompts VLMs for reasons behind a particular model decision in zero-shot and few-shot settings. Here the VLM acts as a simulator that predicts the original model output and explains its prediction. Another technique (NLX-GPT) finetunes an LM to jointly answer and generate a corresponding natural language explanation in a multitask setting. NLX-GPT is completely unrelated to our process as we explicitly focus on “post-hoc” NLEs.  Our proposed model CAuSE tackles this challenge by leveraging the internal hidden state of the multimodal pre-trained classifier encoder to generate a natural language explanation for the classifier decisions. This is the reason, we can compare our proposed model only against the LLM/VLM prompting strategies which also generate post-hoc explanations.
>
> ## References
> [1]  Sundararajan et. al Axiomatic Attribution for Deep Networks. Proceedings of the 34 th International Conference on Machine
> Learning, Sydney, Australia, PMLR 70, 2017.
>
> [2] Bandyopadhyay et al. 2024. Semantify: Unveiling memes with robust interpretability beyond input attribution. In Proceedings of the Thirty-Third International Joint Conference on Artificial Intelligence, IJCAI-24, pages 6189–6197.
>
> [3] Sammani et al. NLX-GPT: A Model for Natural Language Explanations in Vision and Vision-Language Tasks, CVPR 2022

---

> > ### Comment · Reviewer_s4tx · 2024-11-26
> >
> > Thank you for the response. Having read the response and the other discussions on this page, I decide to maintain my original score.

---

### Official Review · Reviewer_cVSQ · 2024-11-03

**Soundness:** 2
**Presentation:** 1
**Contribution:** 2
**Rating:** 3
**Confidence:** 4

**Summary:**

The paper introduces the CAuSE framework for generating post-hoc natural language explanations for multimodal classifiers by leveraging causal abstraction. The proposed method aims to provide causally faithful explanations, evaluated using the proposed Counterfactual F1 score. Experiments are conducted on two visual-textual classification tasks:e-SNLI-VE and Facebook Hateful Memes.

**Strengths:**

- The paper investigates an important area of generating faithful natural language explanations for visual language models.
- The paper proposes a causality-based approach for improving the faithfulness of explanations.

**Weaknesses:**

- The paper does not adequately validate the faithfulness of the generated explanations. More thorough investigation is needed to support the claims.
- The evaluation metric is not sufficient to validate the faithfulness of explanations.
- The paper lacks motivation for design choices in several components of the proposed method.
- The paper suffers from unclear explanations of key components, making it difficult to follow some methodological details.

**Questions:**

- Does the post-hoc classifier need to be trained from scratch with the language models, or can this method be used to interpret pre-trained models?
- In Figure 1, the multimodal encoder appears not to take any text input, which contradicts the description in Section 2.1.
- Why are two language models needed? What is the purpose behind training an LM to reconstruct the input text? Although discussed in ablation studies, the explanation feels rather post-hoc.
- In Equations (2) and (3), is using only the previous token sufficient for next token prediction?
- According to the definition, x_{i-1} should be the ground-truth token, which does not match the autoregressive generation scheme depicted in Figure 1.
- Does sum pooling offer advantages over average pooling, particularly when text lengths vary significantly between samples?
- Is the training conducted end-to-end in a single phase? The training process should be described more clearly.
- How is c_hat from Section 3.1 utilized in this work?
- Counterfactual F1 is not explicitly referenced in any table.
- In Algorithm 1, how is the return value for "Calculate Counterfactual F1 score" determined? The origin of F1 and the definition of the score function are unclear.
- How does the causal abstraction in this method address the limitations mentioned in the Introduction, such as "models injecting their own biases and opinions"? Are there any experimental results supporting the faithfulness of the generated explanations, especially
when the model makes incorrect decisions?
- How are baseline methods generating natural language explanations? Did you include groundtruth explanations in the few-shot prompts? Additionally, the paper would benefit from comparing to more sophisticated baselines, such as Chain-of-Thought (CoT) prompting, to better elicit the decision-making process.

---

> ### Author Response · Authors · 2024-11-19
> **Response to weaknesses and questions**
>
> # Response to Weaknesses
>
> > The paper does not adequately validate the faithfulness of the generated explanations. More thorough investigation is needed to support the claims.
>
> The faithfulness of the generated explanation stems from the fact that LLM machinery is a causal abstraction of the pre-trained multimodal classifier. Further, the fact that LLM machinery is a causal abstraction of the pre-trained multimodal classifier is shown by the high counterfactual F1 score obtained by the CAuSE framework. This validates that the generated explanations are causally faithful to the original pretrained multimodal classifier.
>
> > The evaluation metric is not sufficient to validate the faithfulness of explanations.
>
> The evaluation metric that we propose (counterfactual F1 score) validates the faithfulness of the explanations from an empirical standpoint as discussed in the previous point. Further, we validate its faithfulness on two different datasets. Although satisfactory, we agree that the faithfulness of explanations can be viewed from a qualitative standpoint. Specifically, we could focus on self-consistency checks (Madsen et. al.) to validate the faithfulness of explanations, but these self-consistency checks are only applicable to generative models (LLMs/ VLMs) and not to classifiers.
>
>
> > The paper lacks motivation for design choices in several components of the proposed method.
>
> Architecturally, the proposed framework CAuSE is composed of a pre-trained multimodal classifier (which we call a post-hoc classifier) and an LLM machinery coupled with interchange intervention. LLM machinery explains an existing multimodal classifier. Here we have chosen a standard early-fusion-based multimodal classifier (which uses CLIP+MFB underneath)  from the literature. Naturally, we use it as a black box because the CAuSE framework only needs the hidden representation from the encoder of this multimodal model. Consequently, we do not adequately explain the design choices internal to the multimodal classifier (which we borrow from related literature), which is outside the scope of the proposed framework CAuSE. Other than that, we adequately explain the architecture details in Section 2 (CAuSE framework) and training details in Section 3 (Linguistic Infusion and Interchange Intervention Training).

---

> > ### Author Response · Authors · 2024-11-19
> > **Response to questions**
> >
> > # Response to questions
> >
> > 1. The post hoc classifier explicitly refers to a pre-trained multimodal classifier model (post-hoc classifier). The CAuSE framework is used for post-hoc natural language explanation generation of pre-trained models during inference.
> >
> > 2. The encoder of the post-hoc classifier is multimodal. This means that it takes both image and text inputs, representations for which are then fused before it is passed to the feed-forward neural networks (FFNs). In Figure 1, to avoid redundancy, we only showed the meme input, which consists of an image and text embedded in it. We will make the necessary changes to Figure 1.
> >
> > 3. The reconstruction of the content by the second language model ($\phi_1$) enriches the representation quality of c by forcing the model to learn content-specific information. This is also validated by experiments and elaborated in the Ablation Studies in lines 416 - 418.
> >
> > 4. Equations 2 and 3 are the Causal Language Modelling Loss which is used for next token prediction for autoregressive models like GPT 2. Hence, the equation should instead be $log P_{\phi_1}(x_i|x_{i-1}, x_{i-2},...x_0)$
> >
> > 5. Our proposed model is indeed autoregressive. The CLM loss depicted in equations 1 and 2 should be instead $log P_{\phi_1}(x_i|x_i-1, x_{i-2},...x_0)$. We have updated it in the paper.
> >
> > 6. The sum pooling or the average pooling are modelling choices involved with the post-hoc classifier multimodal encoder. The post-hoc classifier is not part of our proposed CAuSE framework. Our objective is to interpret this pre-trained model and generate natural language explanations for its decisions (outputs) using our proposed framework.
> >
> > 7. The CAuSE takes the hidden state from the encoder of a pre-trained classifier (post-hoc classifier). Then the entire CAuSE framework is trained in an end-to-end fashion while the post-hoc classifier is in inference mode.
> >
> > 8. $\hat{c}$ is the final representation that is obtained by iteratively updating $c$ using the gradient of the autoencoding loss (line no. 181, 182). This is the final representation that is being split into $c_0$ & $c_1$.
> >
> > 9. The counterfactual F1 score (c-F1) has been referenced in Table 2 along with the number of generations and their harmonic mean.
> >
> > 10. The counterfactual F1 score is the F1 score between the output of the post-hoc classifier and the LLM machinery under counterfactual inputs. In Algorithm 1, The XList refers to the list of outputs from the post-hoc classifier and the ZList refers to the list of outputs from the LLM machinery. F1 score is calculated between these two lists.
> >
> > 11. The counterfactual F1 score measures the agreement between the decision of the post-hoc classifier and the LLM machinery under counterfactual inputs. Concretely speaking, *the result corresponding to the altered counterfactual input to the post-hoc classifier should be the same as that of the LLM machinery if they are a causal abstraction of each other*. This indicates that the LLM machinery does not inject its own biases or opinions into the natural language explanation that it generates corresponding to the post-hoc classifier decision. The results of the cF1 score as mentioned in table 2 experimentally establishes the faithfulness of the generated explanations.
> >
> >
> > 12. The baseline VLMs have only been provided with the input text and image in 0-shot setting. They are also provided with a natural language explanation of the post-hoc classifier decision in a few-shot setting when the predicted classifier output matches the true label. In a situation where the post-hoc classifier’s output class is different from the true class label, the ground truth explanation is not provided to the LLM, instead, it is simply mentioned that the “This input is {predicted_class}”.

---

> > > ### Comment · Reviewer_cVSQ · 2024-12-02
> > >
> > > Thank you for addressing my questions.
> > >
> > > After reviewing the responses and discussions, I have decided to maintain my original score due to remaining concerns about the justification for design choices and insufficient validation of the proposed method.

---

### Official Review · Reviewer_4JtQ · 2024-11-09

**Soundness:** 2
**Presentation:** 2
**Contribution:** 2
**Rating:** 5
**Confidence:** 3

**Summary:**

The paper presents a novel framework to produce explanations for multimodal classifiers. The framework addresses several shortcomings of existing methods, aiming to produce natural language, causally faithful, and task-agnostic explanations. The authors provide theoretical guarantees on causal abstraction of their framework under specific assumptions. The authors introduce the Counterfactual F1 score as a proxy for causal faithfulness in multimodal classifier's explanations. The paper includes empirical comparisons to VLM-based baseline methods and justification for different components in their novel loss function, using well-established datasets e-SNLI and Facebook Hateful Memes.

**Strengths:**

+ The paper introduces a novel loss function to jointly train for generating natural language explanations and classifying with causal guarantees.
+ The authors provide theoretical results under specific assumptions that guarantee causal abstraction between the classifier we want to explain and the machinery used to generate faithful natural language explanation.
+ The authors provide reasonable baseline comparisons to VLM-based methods.
+ The paper includes empirical validation for the need for multiple components in the loss function to generate coherent and faithful natural language explanations.
+ The authors include case studies for both when the framework succeeds and fails to provide coherent and faithful explanations, giving some insights into the capabilities and drawbacks of the CAuSE framework.
+ The architecture diagrams provide helpful intuition on how the CAuSE framework works.

**Weaknesses:**

+ The assumptions for Theorem 1 and 2 are restrictive and specifically require that the classifiers $\mathcal{C}_1$ and $\mathcal{C}_2$ share identical weights.
+ The Counterfactual F1 score only accounts for the classifier $\mathcal{C}_2$, without assessing the causal faithfulness of the LLM machinery $\phi_2$ used to generate explanations. As a result, the conclusion drawn from the Counterfactual F1 score between the CAuSE framework and VLM-based baselines is unclear, making it difficult to determine if CAuSE achieves state-of-the-art results in causal faithfulness.
+ Some architecture designs and choices for loss function components are insufficiently justified. (See Questions for questions about the motivation of specific choices)
+ The authors identify scaleability as a challenge in prior methods (e.g., Amnesic Probing), but do not discuss scalability for the CAuSE framework.
+ The paper’s structure hinders linear reading, with multiple references to sections and tables that appear much later. Additionally, the distance between when a choice is introduced and when it is justified makes it challenging to follow.
+ Figures 1 and 2 would benefit from larger font sizes and detailed captions to enhance readability and comprehension of the CAuSE framework.

**Questions:**

+ Line 103: Other than for practical reasons, what is the theoretical motivation for $\phi_1$ and $\phi_2$ to share the same weights?
+ Line 106: How exactly is representation $c$ broken into two components $c_0$ and $c_1$?
+ Line 152, 255, and 272: What motivates the need for both the cross entropy loss $L_C$ and the teacher-student loss $L_{TS}$ in the loss function
+ Line 182: Where does the linguistically infused representation $\hat{c}$ is used in the CAuSE framework?
+ Line 305: Given that Theorem 2 includes the LLM machinery, does the Counterfactual F1 score include the LLM machinery $\phi_2$? If not, how may one include $\phi_2$ in the Counterfactual F1 score, and how may it impact the causal faithfulness measure of the CAuSE framework?
+ Table 2: Why does CAuSE framework often produce gibberish text when provided with counterfactual inputs and some components of the loss function are removed? Is this problem present in the VLM-based baselines?
+ Section 4.2.2: Are there some statistics on how common these errors tend to occur in CAuSE?
+ Appendix C: What is the motivation for restricting the VLM explanation to one sentence?

---

> ### Author Response · Authors · 2024-11-19
> **Response to weaknesses and questions**
>
> Thanks for the review. That was very helpful in uplifting the quality of our paper. Below is our response.
>
> # Response to weaknesses
>
> > The assumptions for Theorem 1 and 2 are restrictive and specifically require that the classifiers and share identical weights.
>
> Please see lines 270-274 and Equation 8. The regularization term $∥W_\mathcal{C1} − W_\mathcal{C2} ∥_F$ explicitly enforces the constraint that the weights of classifiers remain identical as training progresses.
>
> > The Counterfactual F1 score only accounts for the classifier, without assessing the causal faithfulness of the LLM machinery used to generate explanations.
>
> The counterfactual F1 score also considers the LLM machinery ($F$). In line no. 320, $C_2(z')$ should be instead written as $C_2(F(z'))$ and line no. 321 should be read as $C_1(E(x'))$. Also, line no 308 should be read as: $C_1(E(x)) \neq C_1(E(x′))$. We apologize for this inconvenience, and would like to assure you to correct these typos in the paper.
>
>
> > The authors identify scaleability as a challenge in prior methods (e.g., Amnesic Probing), but do not discuss scalability for the CAuSE framework.
>
> The scalability of the CAuSE framework is driven by its use of standard deep learning architectures (LLM and FFN) and its efficient computation of the counterfactual F1 (c-F1) score, requiring only a single forward pass per instance. In contrast, methods like Amnesic Probing, which scale linearly with the number of concepts by projecting onto nullspaces, are computationally intensive and limited to a fixed set of concepts. Thus, CAuSE provides a more scalable approach for assessing concept relevance in multimodal contexts.
>
> We acknowledge the paper's readability can be improved, particularly by clarifying the following point: We should emphasize in Section 3 that we are not proposing any new classifier, rather our task is to explain (via NLE) an already trained classifier through an LLM machinery. Since the design was adopted from the prior literature, its motivation is not explicitly explained.
>
> # Response to questions
>
> 1. We decided that through empirical verification. Particularly, we ran the experiment where $\phi_1$ and $\phi_2$ do not share weights, but the gain in F1 score was minuscule compared to $\phi_1$ and $\phi_2$ sharing weights (which, in turn, leads to lower parameter counts and faster training).
>
> 2. They are broken by passing $c$ through two separate FFNs. It is mentioned in line 107 and 108.
>
> 3. Cross entropy loss is used such that the label is matched between $\mathcal{C}_2$ and $\mathcal{C}_1$, and Teacher-student loss is used such that the probability distribution of the label distribution between $\mathcal{C}_2$ and $\mathcal{C}_1$ matches.
>
> 4. In the CAuSE framework, $c$ is broken down to $c_0$ and $c_1$ which are then used by the respective language models $\phi_0$ and $\phi_1$. Before breaking $c$ into $c_0$ and $c_1$, we perform the linguistic infusion, where $c$ is updated into $\hat{c}$ and then $\hat{c}$ is broken down into $c_0$ and $c_1$ as performed in the code.
>
> 5. Calculating the counterfactual F1 score does require LLM machinery. As addressed to your weakness point: “*The Counterfactual F1 score only accounts for the classifier*”, the algorithm written for calculating the counterfactual F1 scores has typos in lines nos, 308, and 320, 321. Loosely speaking, we pass the ‘to-be’ counterfactual $z'$ through LLM machinery and $\mathcal{C}_2$ to assess whether $z'$ is a true counterfactual for the LLM machinery or not.
>
> 6. Thanks for this question. CAuSE without the IIT loss implies that the LLM machinery is not a causal abstraction of the multimodal classifier. Consequently, which is a counterfactual representation for the multimodal classifier may not be a proper counterfactual representation to the LLM machinery. That is the reason, without IIT loss, when the counterfactual for the multimodal classifier is given to the LLM machinery, it produces gibberish text.
> This problem does not apply to VLM-based baselines as they did not require the hidden state of the multimodal classifier for training using standard causal language modeling (CLM) objective.
>
> 7. Upon manual inspection, these errors do not seem common across the test samples. However, the error associated with a *lack of object-level representation* is more common than the other errors.
>
> 8. Both datasets' ground truth explanations are compact and typically fit in a single sentence. To match this, the VLM was specifically instructed to keep the explanation to one line, as typically VLMs generate verbose explanations if no restriction on generation length is specified in the prompt.

---

> > ### Comment · Reviewer_4JtQ · 2024-11-25
> >
> > I’d like to thank the authors for their work on the rebuttal and clarifications to my questions. I maintain my concern about the lack of justification for key design choices, and there is still a way to go for the general readability and structural order of the paper. After reading everything on this page, I’d like to keep my score.

---

### Author Response · Authors · 2024-11-20
**Gentle reminder to check author response**

Dear reviewers,

Thank you again for your constructive feedback. We have carefully addressed all of your queries and provided detailed responses. Kindly take a moment to review the updates and let us know if you have any further questions or comments.

---

### Meta-Review · Area_Chair_ngFB · 2024-12-21

**Metareview:**

This paper seeks to develop a posthoc explanation of multimodal classifiers based on causality and faithfulness. The reviewers raised several concerns including the strength of the conditions needed for the theorems, the complexity of the approach, a lack of justification of several design choices to name a few. The reviewers were through and followed up after the author response, which did not change their opinion.

**Additional Comments On Reviewer Discussion:**

None beyond what is available in the metareview.

---

### Decision · Program_Chairs · 2025-01-22

Reject